# Essential Genes Discovery in Microorganisms by Transposon-Directed Sequencing (Tn-Seq): Experimental Approaches, Major Goals, and Future Perspectives

**DOI:** 10.3390/ijms252011298

**Published:** 2024-10-21

**Authors:** Gemma Fernández-García, Paula Valdés-Chiara, Patricia Villazán-Gamonal, Sergio Alonso-Fernández, Angel Manteca

**Affiliations:** Department of Functional Biology, Microbiology Area, IUOPA and ISPA, Faculty of Medicine, Universidad de Oviedo, 33006 Oviedo, Spain

**Keywords:** Tn-Seq, transposon, essential genes

## Abstract

Essential genes are crucial for microbial viability, playing key roles in both the primary and secondary metabolism. Since mutations in these genes can threaten organism viability, identifying them is challenging. Conditionally essential genes are required only under specific conditions and are important for functions such as virulence, immunity, stress survival, and antibiotic resistance. Transposon-directed sequencing (Tn-Seq) has emerged as a powerful method for identifying both essential and conditionally essential genes. In this review, we explored Tn-Seq workflows, focusing on eubacterial species and some yeast species. A comparison of 14 eubacteria species revealed 133 conserved essential genes, including those involved in cell division (e.g., *ftsA*, *ftsZ*), DNA replication (e.g., *dnaA*, *dnaE*), ribosomal function, cell wall synthesis (e.g., *murB*, *murC*), and amino acid synthesis (e.g., *alaS*, *argS*). Many other essential genes lack clear orthologues across different microorganisms, making them specific to each organism studied. Conditionally essential genes were identified in 18 bacterial species grown under various conditions, but their conservation was low, reflecting dependence on specific environments and microorganisms. Advances in Tn-Seq are expected to reveal more essential genes in the near future, deepening our understanding of microbial biology and enhancing our ability to manipulate microbial growth, as well as both the primary and secondary metabolism.

## 1. Introduction

### 1.1. The Biological and Biotechnological Relevance of Essential Genes

Essential genes are critical for the survival, growth, and reproduction of all organisms. They encode proteins involved in key cellular processes, without which bacteria cannot sustain life. These processes include DNA replication and repair, RNA transcription, protein synthesis, cell wall biosynthesis, cell membrane integrity and function, central metabolic pathways (such as glycolysis, the TCA cycle, amino acid synthesis, and nucleotide synthesis), cell division, etc. Understanding a microorganism’s essential genes is therefore crucial from a basic research perspective, as it allows us to define its metabolism and biology.

From an applied perspective, essential genes include clinically relevant ones. Several essential genes are directly involved in the production or regulation of virulence factors, such as toxins, adhesins, and enzymes that degrade host tissues [1]. Other essential genes are involved in mechanisms that help bacteria evade the host immune system. For instance, certain essential genes in pathogens encode proteins that modify bacterial surface molecules, making it more difficult for host immune cells to recognise and attack the bacteria. Additionally, essential genes related to nutrient acquisition are key to pathogenicity, as they enable bacteria to compete with the host for essential nutrients, such as iron, which is limited within host environments [2]. Understanding the role of essential genes in bacterial pathogenicity is fundamental for developing new antimicrobial therapies. Since these genes are indispensable for bacterial survival, they represent potential targets for antibiotics and other treatments aimed at weakening or killing the pathogen by disrupting its critical functions [3].

Essential genes may also include biotechnologically relevant ones. For example, there are essential genes that modulate secondary metabolism, by increasing the fitness of microorganisms, by boosting precursors of secondary metabolism, or by regulating the expression of secondary metabolic pathways in addition to controlling microbial survival [3]. These essential genes have potential applications in enhancing the production of bioactive compounds in industrial microorganisms. Knowledge of essential genes can also contribute to generating genome-reduced industrial microorganisms in which non-essential genes are removed, leading to a significant enhancement in secondary metabolite production [4].

### 1.2. Detection of Essential Genes in Bacteria Using Transposon-Directed Sequencing (Tn-Seq)

The identification of essential genes relies on the inability to produce viable mutations. Consequently, a practical definition of essential genes would be those for which it is impossible to generate a viable knockout mutation under the tested conditions, as their identification depends on this inability [5,6]. The most widely accepted technique for elucidating the essential genes of a microorganism is Tn-Seq, which will be reviewed in this work. Other strategies for identifying essential genes are based on technologies such as CRISPR/Cas9 or RNAi, although these are more commonly used in eukaryotic cells [7,8]. Nevertheless, there are examples of these strategies applied to prokaryotes, such as the study by Rousset et al., who identified the essential genes of *E. coli* using a CRISPR-dCas9-based system [9].

Tn-Seq combines random mutagenesis by transposons with next-generation sequencing (NGS) techniques [10,11]. The goal of Tn-Seq is to generate a library of mutants in which each cell/mutant exclusively harbours a single transposon insertion in its genome [12]. These mutants are subsequently pooled, and their collective chromosomal DNA is isolated and sequenced using NGS. Through bioinformatic analysis, essential genes can be identified as those that do not tolerate transposon insertions [13], as mutants with disruptions in these genes fail to grow and are consequently absent from the sequenced mutant mixture.

There are two main variants of Tn-Seq: the first focuses on the search for essential genes, meaning those that cannot tolerate mutations; the second targets conditionally essential genes, meaning those that are necessary under certain conditions, such as stress conditions, but not under others. Tn-Seq is relatively novel, and we are still far from fully harnessing the potential of this methodology. To the best of our knowledge, essential and conditionally essential genes have been investigated via Tn-Seq in 24 bacterial species (*Escherichia coli*, *Pseudomonas aeruginosa*, *Ralstonia solanacearum*, *Porphyromonas gingivalis*, *Dickeya dadantii*, *Streptococcus pyogenes*, *Streptococcus pneumoniae*, *Streptococcus agalactiae*, *Rhodobacter sphareoides*, *Rhodopseudomonas palustris*, *Azoarcus olearius*, *Caulobacter crescentus*, *Mycobacterium tuberculosis*, *Salmonella typhimurium*, *Salmonella enterica*, *Burkholderia cenocepacia*, *Staphylococcus aureus*, *Vibrio cholerae*, *Herbaspirillum seropedicae*, *Listeria monocytogenes*, *Phaeobacter inhibens*, *Pseudomonas stutzer*, *Shewanella amazonensis*, *Shewanella oneidensis*) and 3 yeast species (*Pichia pastoris*, *Saccharomyces cerevisiae* and *Schizosaccharomyces pombe*) (Table 1 and Table 2). In the next sections, we provide a comprehensive state of the art of Tn-Seq in microorganisms.

### 1.3. Main Transposons Used for Random Mutagenesis in Microorganisms

Transposons are mobile genetic elements capable of relocating within or between genomes through “copy and paste” or “cut and paste” mechanisms (Figure 1). Transposons are categorised based on their transposition mechanisms. Class I transposons, also known as retrotransposons, require an RNA intermediate to integrate or duplicate themselves within a genome (Figure 1a), whereas Class II elements operate without such an intermediate, using a “cut and paste” mechanism that directly transfers their DNA during mobilisation (Figure 1b) [14]. Transposons used in Tn-Seq experiments are modified to include the transposase outside the transposable region, thereby preventing further mobility once integrated into the chromosome. Typically, transposons are delivered via a suicide plasmid (i.e., one that cannot replicate in the host) or a plasmid with a temperature-sensitive origin of replication [15]. The codon bias of the transposase is usually optimised for the host microorganism, enhancing transposition efficiency.

To the best of our knowledge, all transposons used in bacterial and yeast Tn-Seq analyses (Table 1) belong to Class II, where transposition is facilitated by an enzyme known as transposase. Class II transposons replicate via the so-called “cut and paste” mechanism, in which transposase recognises the inverted repeats at the ends of the transposon and facilitates its integration by identifying a random target sequence in the host genome. The enzyme cleaves the double-stranded DNA at the target site, enabling the insertion of the transposon into the genome [16] (Figure 1b).

The Tn5 and Mariner transposons are the most commonly used in Tn-Seq analyses, due to their simplicity, wide host range, and nearly random and well-characterised insertion nature [10]. However, the Mariner transposon exhibits a strict insertion preference for thymine–adenine (TA) dinucleotide sites [12]. On the other hand, the Tn5 transposon appears to have a slight preference for cytosine–guanine (CG) dinucleotides. A widely used family of transposons is the Himar family, which is derived from the Mariner transposon.

The choice of an appropriate transposon is critical for the development of a robust Tn-Seq methodology. If the chosen transposon exhibits a preference for specific insertion sites, it is essential to ensure that these sites are both abundant and uniformly distributed across the microorganism’s genome. For example, given the slight preference of the Tn5 transposon for CG dinucleotides [13], its use is advantageous in genomes with a high content of these dinucleotides, making it preferable to Mariner transposons [17]. However, the most important factor in choosing a specific transposon for Tn-Seq development in a particular microorganism is the availability of efficient systems to deliver the transposon into the microorganism, thereby generating the large number of mutants required to create a transposon-saturated mutant library [16].

**Table 1 ijms-25-11298-t001:** Tn-Seq methodologies used to identify essential genes in 14 bacterial species and 3 yeast species growing on rich laboratory media. The methodology, microorganism, transposon, number of essential genes identified, and the software used for Tn-Seq analysis are indicated. TSAS 2.0 (Tn-Seq Analysis Software; https://github.com/srimam/TSAS) [17], ESSENTIALS [18], TRANSIT (https://transit.readthedocs.io/en/latest/transit_overview.html) [19], and Tn-Seq Explorer software (https://github.com/sina-cb/Tn-seqExplorer) [20] are specifically designed for Tn-Seq analyses. Bowtie [21], CLC Genomics Workbench (Qiagen Hilden, Germany), MAQ [22], SAMTools (http://www.htslib.org/) [23], HTseq (https://htseq.readthedocs.io/en/latest/) [24], Galaxy (https://usegalaxy.org/) [25], Python (https://www.python.org), R (https://www.R-project.org/), and Perl (https://www.perl.org, accessed on 20 September 2024) are software not specifically designed for Tn-Seq analyses, for which custom scripts are needed.

Methodology	Microorganism	Transposon	Software	Number of Essential Genes	Reference
Type IIs restriction enzymes	*Ralstonia solanacearum*	Mariner	TSAS	465	[26]
	*Streptococcus pneumoniae*	Himar1	Bowtie, CLC, MAQ	247	[27]
	*Streptococcus pyogenes*	Krmit	SAMTools, HTseq	227 and 241 (two strains)	[5]
	*Streptococcus agalactiae*	Himar1	ESSENTIALS	317	[28]
	*Schizosaccharomyces pombe*	Hermes	Perl, Ruby	1258	[29]
	*Herbaspirillum seropedicae*	Mariner	ESSENTIALS	395	[30]
Circle method	*Rhodobacter sphaeroides*	Tn5	TSAS	493	[17]
	*Rhodopseudomonas palustris*	Tn5	Perl	552	[31]
	*Saccharomyces cerevisiae*	MiniDs	CLC	299	[32]
Random primer method	*Azoarcus olearius*	Tn5	Python, Perl, R	616	[33]
	*Caulobacter crescentus*	Tn5	MAQ	480	[34]
	*Escherichia coli*	Tn5	CLC	233	[35]
Sonication and Illumina adapter ligation	*Mycobacterium tuberculosis*	Tn5371	TRANSIT	458	[36]
	*Salmonella typhimurium*	Tn5	MAQ	353	[37]
	*Porphyromonas gingivalis*	Himar1	Galaxy	463	[38]
	*Burkholderia cenocepacia*	Tn5	Tn-Seq Explorer	398	[39]
	*Pichia pastoris*	TcBuster	Bowtie	1086	[40]

**Table 2 ijms-25-11298-t002:** Tn-Seq methodologies used to identify conditionally essential genes in 18 bacteria. The methodology, microorganism, transposon, developmental condition, the number of conditionally essential genes identified, and the software used for Tn-Seq analysis are indicated. TRANSIT (https://transit.readthedocs.io/en/latest/transit_overview.html) [19] and ARTIST software are specifically designed for Tn-Seq analyses (custom ARTIST scripts are available in the supplementary data of Pritchard et al. [41]). Bowtie (https://bowtie-bio.sourceforge.net/index.shtml) [21,42], CLC Genomics Workbench (Qiagen), Galaxy (https://usegalaxy.org/) [25], Fastp (https://github.com/OpenGene/fastp) [43], Python (https://www.python.org), and Perl (https://www.perl.org, accessed on 20 September 2024) are software not specifically designed for Tn-Seq analyses, for which custom scripts are needed.

Methodology	Microorganism	Transposon	Software	Developmental Condition	Number of Essential Genes or Conditional Essential Genes	Reference
Type IIs restriction enzymes	*Dickeya dadantii*	Himar9	TRANSIT	Survival in chicory plants	96	[44]
	*Salmonella enterica*	Tn5	Python	Diluted LB medium, bile acid, high temperature	105	[45]
	*Vibrio cholerae*	Mariner	CLC	Immunity	8	[46]
	*Bacillus subtilis*	Mariner	CLC, Bowtie	Swarming motility	36	[47]
	*Listeria monocytogenes*	Tn5	Fastp, Bowtie2	Low temperature	140	[48]
Sonication and Illumina adapter ligation	*Mycobacterium tuberculosis*	Himar1	TRANSIT	Antibiotic	50	[49]
	*Vibrio cholerae*	Himar1	CLC	Intestinal colonisation	400	[49]
C-tailing	*Escherichia coli*	Mini-Tn10	Galaxy	Antibiotic	140	[50]
	*Pseudomonas aeruginosa*	Himar1	Python	Desiccation	97	[51]
	*Salmonella enterica*	Tn5	ARTIST	Desiccation	61	[52]
	*Streptococcus pneumoniae*	Mariner	Bowtie	Desiccation	42	[53]
	*Staphylococcus aureus*	Mariner	Bowtie	Invasive infection	200	[54]
	*Salmonella typhimurium*	Tn5	TRANSIT	Iron-restriction	336	[55]
Random barcode (RB-Tn-Seq)	*Escherichia coli*	Mariner	Perl	Different Bacterium–carbon source combinations	5196	[56]
	*Phaeobacter inhibens*					
	*Pseudomonas stutzer*					
	*Shewanella amazonensis*					
	*Shewanella oneidensis*					

## 2. Tn-Seq Approaches in Microorganisms

### 2.1. Number of Transposon Insertional Mutants Required for a Saturated Mutant Library

Tn-Seq relies on the creation of a saturated library (i.e., the probability of mutation by transposition of any open reading frame is close to 1) of random mutants. Achieving this saturation is often the rate-limiting step in the process. If an efficient transposition system is not available in the microorganism of interest, it will not be possible to generate the hundreds of thousands of mutants necessary to obtain a saturated library of random mutants [16].

Zhang et al. [57] defined a formula to objectively calculate the number of transposon insertion colonies needed to obtain a saturated library. For this, they used a derivative of Poisson’s law, N = ln(1 − P)/ln(1 − f), where f is the average gene size over the total size of the microorganism’s genome. Furthermore, to ensure that transposon insertions cover approximately 99.99% of the genome, *p* = 0.9999 is used. Thus, the number of transposon mutants required to achieve a saturated transposon mutant library depends on the size of the genome of the microorganism of interest.

### 2.2. Biases in Transposon Mutagenesis

Another factor to consider is that there may be regions in the genome that do not allow insertions. This can occur due to the presence of bound proteins, stable secondary structures, or, in the case of transposons with preferences for specific sequences, a lack of suitable insertion sites. For example, in eukaryotes like the yeasts *Saccharomyces cerevisiae* and *Saccharomyces pombe*, the presence of nucleosomes decreases integration frequencies; only 33% of all insertions occur in open reading frames (ORFs). This is attributed to the tightly folded nature of the ORFs, which are associated with histones, making them less accessible to transposons [13]. To the best of our knowledge, yeast species are the only eukaryotes in which Tn-Seq has been successfully performed [29,32,40].

Furthermore, due to the organisation of the prokaryotic genome into operons (clusters of co-regulated genes with related functions), mutating a gene within an operon can lead to the repression of all downstream genes [58,59]. It is known that essential genes are particularly abundant in operons, so Tn-seq has the potential to incorrectly classify non-essential genes as essential due to polar effects in operons containing a mixture of both essential and non-essential genes [60]. In this context, only the last gene with a fitness defect can be classified as essential due to polar effects. Upstream genes may or may not exhibit a fitness defect when individually mutated and can only be classified as potentially essential.

## 3. Transposon-Directed Sequencing

Once the saturated mutant library has been obtained, the next step is to perform an NGS of the pooled chromosomal DNA from all mutants to identify the transposon insertion sites (transposon-directed sequencing, Tn-Seq). There are different strategies for Tn-Seq, but they all rely on identifying the transposon sequence and sequencing the adjacent chromosomal DNA region, thereby identifying the transposon insertion sites in the chromosome and the interrupted ORFs. The different Tn-Seq variants are mainly distinguished by how the specific amplification of transposon-chromosomal DNA sequences is carried out from the mutant chromosomal DNA mixture [10]. Although various next-generation sequencing methodologies exist, to the best of our knowledge, all published Tn-Seq protocols have utilised Illumina sequencing. The most important published Tn-Seq strategies are detailed in the following sections and summarised in Table 1 and Table 2.

### 3.1. Methods Based on Type IIs Restriction Enzymes

Methods utilizing type IIs restriction enzymes are employed in mutant libraries generated from transposons with recognition sites for these enzymes in the inverted repeats [16]. An example is the Himar1 transposon, which is derived from the Mariner transposon and modified to create a cleavage site for the type IIs endonuclease *Mme*I in the transposon’s inverted repeats [13]. *Mme*I recognises the sequence 5′ TCCRAC 3′ and cuts the DNA 20 base pairs after it, generating 20 base pairs downstream of the transposon inverted repeats, which is long enough to identify the chromosomal transposon integration site. Afterwards, one of the Illumina NGS adapters is added, followed by PCR amplification to incorporate the other Illumina adapter. The resulting fragments are then sequenced by Illumina NGS, and the insertion sites in the genome are mapped, based in the 20-nucelotide tails [12] (Figure 2).

Tn-Seqs based on type IIs restriction enzymes have successfully identified essential genes in *Ralstonia solanacearum* [26], *Streptococcus pneumoniae* [27], *Streptococcus pyogenes* [5], *Streptococcus agalactiae* [28], *Schizosaccharomyces pombe* [29], and *Herbaspirillum seropedicae* [30] (Table 1). A total of 465 genes essential for the growth of *Ralstonia solanacearum* in rich medium were identified by Su et al. [26]. *Van Opijnen* et al. [27] identified 247 essential genes in *Streptococcus pneumoniae*. Later, Le Breton et al. [5] identified 264 essential genes in two pathogenic strains of *Streptococcus pyogenes*, validating the essentiality of some of them by creating conditional expression mutants. A comparison of essential genes between *S. pneumoniae* and *S. pyogenes* revealed significant overlaps, providing valuable insights for the development of new antimicrobials to treat infections by pathogenic streptococci [5]. Hooven et al. [28] found 317 essential genes in *Streptococcus agalactiae*. These essential genes were enriched for fundamental cellular housekeeping functions, such as acyl-tRNA biosynthesis, nucleotide metabolism, and glycolysis. Notably, there was a 93% concordance between the essential genes of *S. agalactiae* and *S. pyogenes* [28]. Guo et al. [29] identified 1258 essential genes in the yeast *Schizosaccharomyces pombe*, including genes involved in cell division, DNA repair, stress response, and meiosis. Finally, Rosconi et al. [30] identified 395 essential genes in the *Herbaspirillum seropedicae* endophyte bacterium when cultured in rich tryptone yeast extract medium.

Tn-Seq based on Type IIs restriction enzymes have also proven effective in identifying conditionally essential genes in *Dickeya dadantii* [44], *Salmonella enterica* [45], *Vibrio cholerae* [46], *Bacillus subtilis* [47], and *Listeria monocytogenes* [48] (Table 2). Royet et al. (2019) identified 96 conditionally essential genes in the *Dickeya dadantii* pathogen involved in chicory plant pathogenesis. Khatiwara et al. [45] reported the identification of 105 conditionally essential genes in *Salmonella enterica* growing in diluted LB medium supplemented with bile acid or at high temperature. Dong et al. [46] identified eight conditionally essential genes related with the immune response against *Vibrio cholerae*. Sánchez et al. [47] discovered 36 conditional genes required for swarming mobility in *Bacillus subtilis*. Finally, Wu et al. [48] found 140 genes essential for growth at low temperatures in *Listeria monocytogenes*.

### 3.2. Circle Method

The circle method was designed by Gallagher et al. [61]. In this strategy, the total chromosomal DNA from a pool of mutants is fragmented by sonication, and an Illumina adapter is attached to all free ends. The sample is then digested with a restriction enzyme that cuts within the transposon near one of its ends. After the selection of the DNA size, the DNA fragments are denatured and circularised through a ligation reaction using the Gene Collector technique [62]. The purpose of using the Gene Collector technique is to selectively eliminate DNA fragments that do not contain the transposon. This technique involves using a primer (collector oligo) that hybridises to both ends of the target sequence, in this case, the Illumina adapter and the region adjacent to the restriction cut site. A thermostable ligase (Ampligase™ Thermostable DNA Ligase, Biosearch Technologies, Tokyo, Japan) is used, and cycles of DNA denaturation at 95 °C are alternated (to generate single-stranded DNA), with annealing cycles at 67 °C to allow the hybridisation of the collector oligo and subsequent ligation of double-stranded DNA (i.e., the DNA where the primer hybridised). If the restriction enzyme used does not leave blunt ends, these should be created before circularisation by using an exonuclease to remove the single-stranded overhangs left by the enzyme (e.g., S1 nuclease, which can degrade single-stranded DNA) or a polymerase (e.g., the Klenow fragment of *E. coli* DNA polymerase) to fill in these single-stranded overhangs. This results in double-stranded DNA with blunt ends, which are essential for the subsequent ligation and circularisation of the fragments. The T4-DNA ligase only catalyses the ligation of double-stranded DNA, which corresponds to the DNA where the collector oligo hybridised; the rest of the DNA remains as non-circularised linear DNA. Non-circularised fragments are then degraded using an exonuclease that degrades linear DNA from its ends but cannot degrade circularised DNA. Subsequently, the transposon–genome junctions of the circularised fragments are amplified by PCR, during which the second Illumina adapter is introduced. Finally, as with other methodologies, the PCR products are sequenced using Illumina technology, and each sequence read is then mapped to the genome (Figure 3).

The circle method has been employed to study the essential genes of *Rhodobacter sphaeroides* [17], *Rhodopseudomonas palustris* [31], and *Saccharomyces cerevisiae* [32]. Burger et al. [17] identified 493 essential genes in *Rhodobacter sphaeroides*, which included numerous genes involved in bacteriochlorophyll biosynthesis and genes encoding pigment-binding proteins of the photosynthetic apparatus that are required for photosynthetic growth [17]. Pechter et al. [31] used transposon mutagenesis and Tn-Seq to identify 552 essential genes required for cell viability in *Rhodopseudomonas palustris* during aerobic growth on semi-rich medium. *Rhodopseudomonas palustris* serves as a model organism for studies of anaerobic aromatic compound degradation, hydrogen gas production, nitrogen fixation, and photosynthesis, and the knowledge of its essential genes can contribute to improve these processes [31]. Michael et al. [32] identified 552 essential genes in the yeast *Saccharomyces cerevisiae*.

### 3.3. The Random Primer Method

In this methodology, an initial PCR is performed using a transposon-specific primer and various primers with random sequences that hybridise at a specific frequency throughout the genome. This process generates amplicons of an average size suitable for NGS using the Illumina platform. Subsequently, the fragments from this first PCR are then amplified with two primers that allow the incorporation of Illumina adapters for NGS [33] (Figure 4).

The random primers method has been employed to identify essential genes in *Azoarcus olearius* [33], *Caulobacter crescentus* [34], and *Escherichia coli* [35]. Harten et al. [33] identified 616 essential genes under nitrogen fixation conditions in the endophyte *Azoarcus olearius* BH72. Most of these essential genes are related to core cellular functions and cell viability [33]. *Caulobacter crescentus* is a model organism for the study of bacterial cell division [34]. Christen et al. [34] studied their essential genes by means of Tn-Seq, identifying 480 essential ORFs, including transcription factors, RNA polymerase sigma factors, and anti–sigma factors. Choe et al. [35] used Tn-Seq to identify 233 of the 301 essential genes previously validated by knockout mutagenesis [63].

### 3.4. Sonication and the Illumina Adapter Ligation Method

The method of random chromosomal DNA fragmentation followed by Illumina adapter ligation is probably the simplest strategy for Tn-Seq library preparation. In this methodology, DNA is randomly fragmented by sonication, Illumina adapter sequences are ligated, and then a PCR is performed using an oligonucleotide that hybridises with part of the transposon to enrich the sample [36] (Figure 5). Although this strategy is effective, like other methodologies that involve random DNA cleavage it often requires an optimisation step and a considerable amount of initial DNA [10].

The method of random chromosomal DNA fragmentation followed by Illumina adapter ligation has been used to identify essential genes in *Mycobacterium tuberculosis* [36], *Salmonella typhimurium* [11,64], *Porphyromonas gingivalis* [38], *Burkholderia cenocepacia* [39], and the yeast *Pichia pastoris* [40] (Table 1). Griffin et al. [65], DeJesus et al. [66], and Minato et al. [36] studied the conditionally essential genes of *Mycobacterium tuberculosis*, the causative agent of tuberculosis. The three reports coincided in the identification of 458 essential genes [36]. Barquist et al. [11] identified 353 ORFs required for growth under laboratory conditions, which closely resemble those also identified previously by Knuth et al. [64] in *Salmonella*. Klein et al. [38] analysed essential genes in the Gram-negative anaerobic bacterium *Porphyromonas gingivalis*, which is associated with periodontal disease, identifying 463 genes essential for viability in vitro. These included genes involved in cell division, lipid transport, and metabolism, as well as translation and ribosome function. Higgins et al. [39] analysed the essential genes in the opportunistic pathogen *Burkholderia cenocepacia*, identifying 398 essential genes in laboratory cultures. Finally, Zhu et al. [40] identified 1086 essential genes in the yeast *Pichia pastoris*.

The method of random chromosomal DNA fragmentation followed by Illumina adapter ligation has also been employed to identify conditionally essential genes in *Mycobacterium tuberculosis* [49] and *Vibrio cholerae* [67]. Carey et al. [49] identified 50 genes related with antibiotic resistance in the pathogen *Mycobacterium tuberculosis*, while Fu et al. [67] identified 400 conditionally essential genes necessary for intestinal colonisation by *Vibrio cholerae*.

### 3.5. C-Tailing-Based Methods

C-tailing-based methods represent one of the most recent additions to the various Tn-Seq techniques. The C-tailing procedure uses the activity of a terminal transferase to add poly-C tails to the 3′ end of DNA. These poly-C tails then act as binding sites for a poly-G primer, allowing the amplification of transposon insertion sequences [10].

In this strategy, two variants can be found. On the one hand, a linear PCR can be performed using a single transposon-specific primer, followed by the addition of the C-tail (Figure 6a–d); on the other hand, C-tails are added to randomly fragmented chromosomal DNA. In both scenarios, the Illumina sequences are introduced in a subsequent exponential PCR using two primers, one hybridising with the transposon and another one hybridising with the poly-C tail [50] (Figure 6e–h).

C-tailing-based Tn-Seq methods allowed the identification of conditionally essential genes in *Escherichia coli* [50], *Pseudomonas aeruginosa* [51], *Salmonella enterica* [52], *Streptococcus pneumoniae* [53], *Staphylococcus aureus* [54], and *Salmonella typhimurium* [55] (Table 2). Shan et al. [50] identified 140 genes necessary for resistance to aminoglycosides in *Escherichia coli*. Genes necessary for resistance under desiccation were found in Pseudomonas aeruginosa (97 genes) [51], *Salmonella enterica* (61 genes) [52], and *Streptococcus pneumoniae* (42 genes) [53]. Wilde et al. [54] identified 200 genes necessary for *Staphylococcus aureus* invasive infection [54]. Additionally, Karash and Kwon [55] identified 336 iron-dependent essential genes in *Salmonella typhimurium*.

### 3.6. Random Barcode Transposon Sequencing (RB-Tn-Seq)

The strategy known as random barcode transposon sequencing (RB-Tn-Seq) is a variant of Tn-Seq that has emerged as a promising solution for analysing multiple conditions simultaneously [13]. This method allows the identification of conditional essential genes, i.e., genes that are essential under different growth conditions.

In this method, random barcodes (20-nucleotide sequences) are incorporated into each transposon before generating the saturated library of random mutants. Subsequently, Tn-Seq sequencing is performed using any of the methods mentioned in the previous sections. In this way, all transposon insertions are identified, and each specific insertion in the genome is correlated with a unique barcode [12]. The analysis of essential genes under different conditions is conducted in subsequent experiments. For this, clones from the saturated mutant library are grown under the specific conditions to be studied. Chromosomal DNA is isolated from the mixture of mutants in each condition. After mixing the DNAs from the different conditions in equal proportions, a single PCR is performed, amplifying the barcodes using universal primers flanking the barcode region. Finally, each unique barcode associated with the insertion in each gene is sequenced and quantified. This methodology allows the detection of conditional essential genes, i.e., genes that disappear under a condition because their mutants cannot grow, as well as genes that, while not essential, show a significantly reduced abundance under particular conditions [13] (Figure 7).

The random barcode transposon sequencing technique was used to create mutant libraries harbouring barcodes in *Escherichia coli*, *Phaeobacter inhibens*, *Pseudomonas stutzer*, *Shewanella amazonensis*, and *Shewanella oneidensis* [56]. These mutant libraries allowed the identification of 5,196 genes necessary under different carbon sources in these bacteria [56].

## 4. Identification of Transposon Insertion Sites Through Bioinformatics Analysis

Once the NGS Tn-Seq raw data are obtained, bioinformatics analyses are necessary for identifying essential genes. A typical analysis can be summarised as follows: after obtaining the sequencing reads, the Illumina adapter sequences are removed, sequences containing the transposon are filtered out, and the sequences adjacent to the transposon are aligned against the reference genome to pinpoint the exact insertion sites of the transposons. While some researchers prefer to develop their own scripts to carry out the data analysis, others opt to use various software specifically designed for identifying essential genes from Tn-Seq data. Although these software tools are quite similar, they differ in details such as the preprocessing and alignment of reads, normalisation techniques, and the statistical models or tests employed [12].

### 4.1. Tn-Seq Analysis Software (TSAS)

The TSAS (Tn-Seq Analysis Software) was developed by Burger et al. in 2017 [17] and has proven successful in determining the essential genes of *Rhodobacter sphaeroides* [17] and *Ralstonia solanacearum* [26]. Prior to inputting the sequences into the software, a processing step is required. First, sequences containing the transposon are identified. Then, the transposon and adapter sequences are removed, and the remaining sequences are aligned against the reference genome, allowing for no or few base mismatches [17,26]. This alignment file, along with the organism’s genome sequence in FASTA format and the genomic positions where the insertions occurred, must be provided to the tool. These files are used to assign reads and insertion sites to the corresponding genes, establishing the basis for the subsequent statistical analysis.

One of the advantages of the TSAS software is its ability to address common issues encountered during Tn-Seq analysis, including biases in library construction, PCR amplification biases and non-random transposon insertions. To counteract these potential biases, a read-capping procedure can be employed to set a maximum limit on the number of reads allowed at a given insertion site [17].

The current version of TSAS is TSAS 2.0, written in Python (https://github.com/srimam/TSAS; accessed on 20 September 2024).

### 4.2. TRANSIT Software

The TRANSIT software (https://transit.readthedocs.io/en/latest/transit_overview.html; accessed on 20 September 2024) was developed by DeJesus et al. in 2015 [19] for Tn-Seq analysis from transposon random mutant libraries specifically created using Himar1. The TRANSIT software has been widely used in studies to determine essential genes in various bacterial species, such as *Dickeya dadantii* [44], *Mycobacterium tuberculosis* [36,49], and *Salmonella typhimurium* [55]. Similar to other Tn-Seq analysis tools, the sequences must first undergo preprocessing, which involves identifying sequences carrying the transposon and removing both transposon and adapter sequences. Subsequently, sequences are size-selected and mapped to the reference genome. The aligned sequences are input into TRANSIT, which performs various statistical analyses to determine the essentiality of genes [36,44,55]. One of the significant advantages of TRANSIT is its ability to examine the genome for essential regions or domains, rather than focusing only on complete genes. This allows for a more detailed analysis of genetic essentiality, identifying critical functional elements within genes [13].

### 4.3. ESSENTIALS Software

ESSENTIALS is an open-source, web-based software tool widely used for Tn-Seq analysis, developed in 2012 by Zomer et al. [18]. The ESSENTIALS software has been successfully employed to determine essential genes in *Herbaspirillum seropedicae* [30] and *Streptococcus agalactiae* [28]. ESSENTIALS requires sequence preprocessing, alignment to a reference genome, and subsequent statistical analysis to identify essential genes. The software is noted for its streamlined interface and ease of use. Unfortunately, the links to access ESSENTIALS (http://bamics2.cmbi.ru.nl/websoftware/essentials/ and http://trac.nbic.nl/essentials/) provided by Zomer et al. [18] do not appear to be working (attempted access on 20 September 2024).

### 4.4. ARTIST Software

The ARTIST pipeline was developed in 2014 by Pritchard et al. [41] to study conditionally essential genes. ARTIST uses simulation-based normalisation to compensate for experimental noise in Tn-Seq analyses, optimising the identification of these genes. The ARTIST pipeline has been used to identify conditionally essential genes under desiccation in *Salmonella enterica* [52]. ARTIST was developed to analyse Tn-Seq datasets generated using Mariner-based transposons, which insert specifically at TA dinucleotides, but it should be adaptable to Tn-Seq analysis using Tn5 transposons, which have no sequence specificity [41].

### 4.5. Tn-Seq Analysis Using R, Perl, Ruby, and Python Programming Lenguages

Most authors have chosen to develop their own scripts in programming languages such as R, Perl, Ruby, and Python to perform essential gene analysis (Table 1 and Table 2). As with the other approaches described above, the sequences are first filtered based on the presence of the transposon, and then sequences originating from the transposon and Illumina adapters are removed. Next, the remaining sequences are aligned with the reference genome to achieve a perfect match of a specified number of base pairs (around 15 bp) from the transposon’s end. The transposon insertion site in the genome is calculated using the first base that aligns with the reference genome.

Subsequently, specific scripts are created according to the needs of the particular studies. For example, in the analyses of *Caulobacter crescentus* [34] and *Azoarcus olearius* [33], where the Tn5 transposon was used, it was necessary to include a correction in the script for the insertion site, considering the 9 bp duplication that this transposon produces upon insertion in the genome. The essential gene analysis is carried out by assigning the insertion sites to an ORF. Insertions in the +1 reading frame of an ORF were discarded, as the transposon used in these studies [33,34] had a promoter and a start codon (ATG), which results in the formation of an mRNA from the ORF when the transposon inserts into the +1 reading frame. Additionally, insertions located at the stop codon were also discarded, as well as those located in the first and last 9 bp, again considering the 9 bp duplication caused by the transposon. Genes whose ORFs lack any insertion sites are defined as genes not targeted by the transposon and, therefore, potentially essential. The authors also considered essential those genes with insertions only in the first 10% of the ORF or with 80% of the ORF without insertions. In other words, insertion bias within an ORF indicates essentiality, as non-essential genes should have insertions throughout the entire ORF [33,34].

To analyse potentially essential consecutive genes on the chromosome, the probability that n essential genes are clustered together on the chromosome must be calculated. To do this, a permutation test of all ORFs sequenced in the Tn-Seq with transposon insertions (non-essential) and without insertions (possibly essential) is performed. The probability that n essential genes are randomly grouped in the genome is calculated using a 100,000-simulation permutation test. This test determined that, in these bacteria, regions of six or fewer consecutive essential genes could occur by pure chance, meaning regions with six or more genes without transposon insertions indicate possible clusters of essential genes whose collective position on the chromosome is not random and reflects some essential biological function [33,34].

## 5. Essential and Conditionally Essential Genes in Bacteria Identified by Tn-Seq

### 5.1. Essential Genes in Bacteria

As indicated above, essential genes have been analysed using different Tn-Seq workflows in at least 14 bacteria grown in nutrient-rich laboratory culture media (Table 1). A comparative analysis of gene orthologues reveals a conserved core of 133 essential genes shared across these bacteria (Figure 8). These genes encode important cell division proteins (*ftsA*, *ftsH*, *ftsW*, *ftsY*, *ftsZ*), DNA replication proteins (*dnaA*, *dnaE*, *dnaG*), ribosomal proteins (*rplB*, *rplC*, *rplD*, *rplF*, *rplJ*, *rplK*, *rplL*, *rplM*, *rplN*, *rplO*, *rplQ*, *rplR*, *rplT*, *rplU*, *rplV*, *rplW*, *rplX*, *rpmD*, *rpoA*, *rpoB*, *rpoC*, *rpoD*, *rpsA*, *rpsB*, *rpsC*, *rpsD*, *rpsE*, *rpsG*, *rpsH*, *rpsI*, *rpsJ*, *rpsK*, *rpsL*, *rpsM*, *rpsN*, *rpsQ*), enzymes involved in cell wall synthesis (*murB*, *murC*, *murD*, *murE*, *murF*, *murG*), and enzymes related to amino acid synthesis (*alaS*, *argS*, *aspS*, *glyQ*, *glyS*, *hisS*, *ileS*, *lysS*, *metK*, *serS*, *thrS*, *tyrS*, *valS*) (Figure 8).

Only three genes (*dnaE, gyrB, rpoB*) have been identified as essential in all 14 bacteria included in Figure 8. This result seems to indicate that the vast taxonomic and metabolic diversity present in prokaryotes leads to exceptions where most of these 133 essential genes can be mutated, resulting in a viable phenotype.

According to the Gene Ontology database (GO), these essential genes are highly enriched in GO protein classes related to translational proteins (GO accession PC00263) and metabolite interconversion enzymes (GO accession PC00262) (Figure 9a). Additionally, the catalytic activity (accession GO: 0003824), binding (accession GO: 0005488) and structural molecule activity (GO: 0045182) GO molecular functions are highly represented between these essential genes (Figure 9b).

### 5.2. Conditionally Essential Genes in Bacteria

Conditionally essential genes were analysed using different Tn-Seq workflows in at least 18 bacteria grown under different conditions (Table 2). Fifty-two conditionally essential genes were identified as being shared among at least two of these bacteria (Figure 10a,b). Unlike the essential genes, which were highly conserved between the different bacteria (Figure 8), the conservation of conditionally essential genes was much lower (Figure 10a,b), indicating that conditionally essential genes are variable and depend on the specific developmental conditions and microorganisms. In this regard, the overlap between conditionally essential genes and essential genes is very low (Figure 10c).

According to the Gene Ontology database, these conditionally essential genes are highly enriched in metabolite interconversion enzymes (GO accession PC00264) (Figure 11a). The most prevalent GO molecular function among these genes is catalytic activity (accession GO::0003824) (Figure 11b).

## 6. Conclusions and Future Perspectives

Transposon-directed sequencing has become crucial for identifying essential genes in microorganisms. To date, nine transposon types have been used to create saturated random mutant libraries, with up to six strategies developed to enrich chromosomal DNA sequences harbouring transposons. These strategies rely on PCR with transposon-binding primers and include methods such as type IIs restriction enzyme-based techniques, the circle method, random primer methods, sonication with Illumina adapter ligation, C-tailing, and random barcode transposon sequencing. To the best of our knowledge, all Tn-Seq experiments have utilised the Illumina NGS platform, and the need to attach Illumina adapters to the sequences represents a major restriction in the design of Tn-Seq strategies. The adoption of alternative NGS technologies, such as nanopore sequencing, is expected to facilitate and expand Tn-Seq analyses in the near future.

Despite over 20 years of research describing the essential genomes of at least 14 bacteria and 3 yeast species, as well as conditionally essential genes in 18 bacteria, many bacterial taxa remain unexplored. A core of 133 essential genes is highly conserved across these bacteria, primarily involved in fundamental cellular pathways such as cell division, protein synthesis, cell wall formation, and amino acid synthesis. To our knowledge, no study has previously compared the essential genes of as many as 14 bacteria, making this list of 133 essential genes highly valuable. Many essential genes lack clear orthologues across different microorganisms, suggesting that these genes may be unique to the specific microorganism under investigation. These essential genes, along with conditionally essential genes, could be crucial for understanding the biology of pathogenic microorganisms (e.g., identifying new therapeutic targets) or industrial microorganisms (e.g., identifying genes modulating secondary metabolism). Despite the relatively extensive data on essential and conditionally essential genes, few studies have explored their significance.

Recent analyses of random mutant libraries generated by transposons have designed workflows beyond traditional growth-based selection approaches, enabling the identification of genes involved in important cellular processes (reviewed in Cain et al. 2020 [12]). For example, motility genes were identified in *Pseudomonas aeruginosa* and *Escherichia coli* by “racing” random mutant libraries on agar plates and sequencing the transposon insertion sites of cells that grew outside the main population (more motile) and inside the main population (less motile) [68,69]. Additionally, the separation of *Klebsiella pneumoniae* random mutants by density gradients, followed by Tn-Seq, allowed for the identification of genes involved in capsule production [70]. In another example, random mutants were separated by cell sorting based on fluorescence using DNA-binding stains, enabling the identification of efflux pump genes in *Mycobacterium tuberculosis* [71]. Additionally, the combination of microfluidics with the encapsulation of *Streptococcus pneumoniae* transposon mutants in growth-medium-in-oil droplets, facilitating individual growth, showed that 1–3% of *Streptococcus pneumoniae* mutants exhibited an altered fitness when grown in isolation [72]. It is expected that, in the near future, new methodologies such as single-cell techniques [73] and others will be applicable to libraries of random mutants generated by transposons, allowing for the extraction of much more information from these libraries. For example, some studies have successfully explored the use of promoters within transposons to enable the overexpression or repression of downstream genes, revealing phenotypes not evident from gene disruption alone [34,74,75,76].

Given these advances in Tn-Seq methodologies, it is expected that the essential genes of many more microorganisms will be discovered in the near future. This progress will significantly deepen our understanding of microbial biology and improve our ability to manipulate microbial growth and metabolism for both therapeutic and industrial applications.

## Figures and Tables

**Figure 1 ijms-25-11298-f001:**
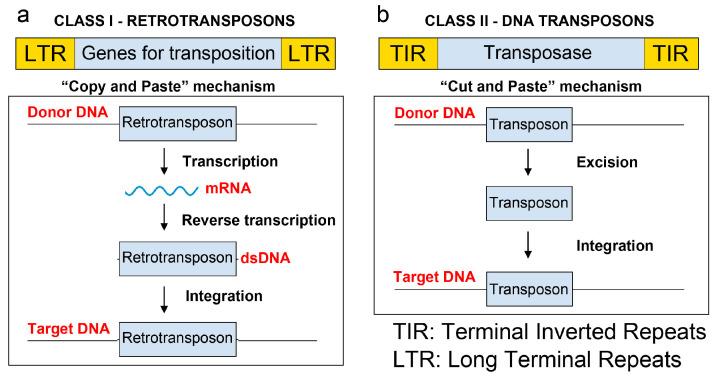
Outline of the two classes of transposable elements. (**a**) Retrotransposons (Class I). (**b**) DNA transposons (Class II). Upper panels illustrate the most common retrotransposon and transposon types. Lower panels illustrate the “copy and paste” and “cut and paste” mechanisms.

**Figure 2 ijms-25-11298-f002:**
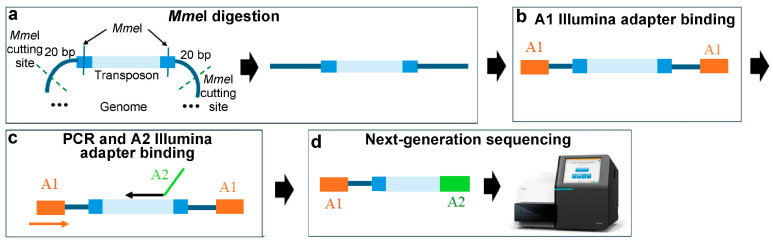
Tn-Seq procedure based on type IIs restriction enzymes. (**a**) *Mme*I digestion. (**b**) A1 Illumina adapter binding. (**c**) PCR and A2 Illumina adapter binding (arrows indicate PCR primers). (**d**) NGS. The *Mme*I recognition site is located within the inverted transposon repeats. *Mme*I cuts 20 bp downstream from the two terminal inverted repeats, releasing the transposon with 20 nucleotides of chromosomal DNA at each end. The transposon sequence is shown in blue, inverted repeats in dark blue; Illumina adapter 1 (A1) is in orange; Illumina adapter 2 (A2) is in green.

**Figure 3 ijms-25-11298-f003:**
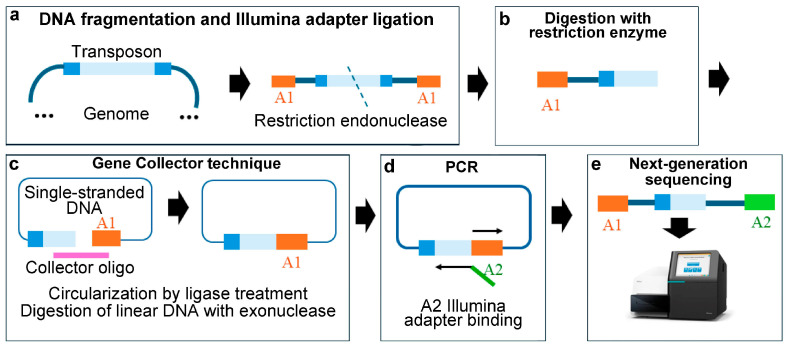
Tn-Seq procedure based on the circle method. (**a**) The DNA from the mutant library is randomly fragmented, and Illumina adapter A1 is ligated to both ends. (**b**) Digestion with a restriction enzyme that cuts within the transposon. (**c**) Selection of DNA fragments containing the transposon using the Gene Collector technique, which includes DNA ligation and the digestion of single-stranded DNA with an exonuclease [61]. (**d**) PCR and addition of the A2 Illumina adapter. (**e**) Next-generation sequencing. The transposon is shown in blue, with its inverted repeats in dark blue; Illumina sequencing adapter 1 (A1) is in orange; adapter 2 (A2) is in green.

**Figure 4 ijms-25-11298-f004:**
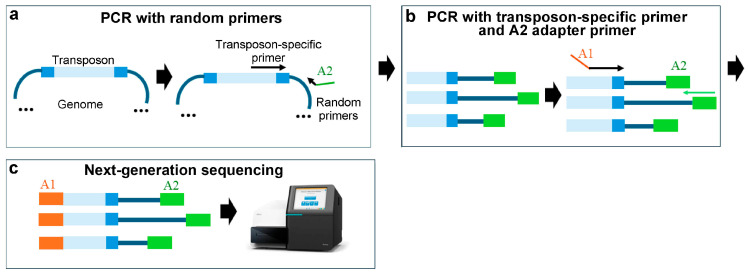
Tn-Seq procedure based on PCR with random primers. (**a**) A first PCR is performed using a transposon-specific primer and various random primers that incorporate the Illumina adapter 2 sequence. (**b**) A second PCR is carried out with a primer that incorporates Illumina adapter 1 and another primer that hybridises with adapter 2. (**c**) Next-generation sequencing. Adapter 1 (A1) is shown in orange; adapter 2 (A2) is shown in green; the Tn5 transposon is shown in blue, with the inverted repeats in dark blue.

**Figure 5 ijms-25-11298-f005:**
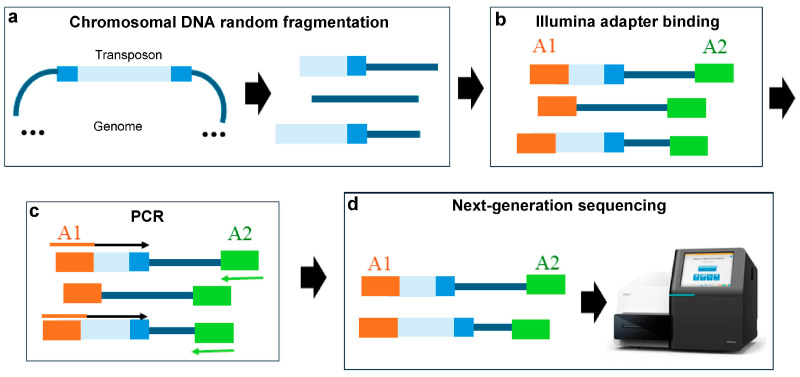
Tn-Seq procedure based on sonication and Illumina adapter ligation. (**a**) The DNA from the mutant library is randomly fragmented by sonication. (**b**) Both Illumina adapters are ligated to the fragments. (**c**) A PCR is performed with a primer that hybridises with adapter 1 and part of the transposon sequence, and with a primer that hybridises with adapter 2. Arrows indicate PCR primers. (**d**) Next-generation sequencing. The transposon is shown in blue, with its inverted repeats in dark blue; Illumina adapter 1 (A1) is shown in orange; Illumina adapter 2 (A2) is shown in green.

**Figure 6 ijms-25-11298-f006:**
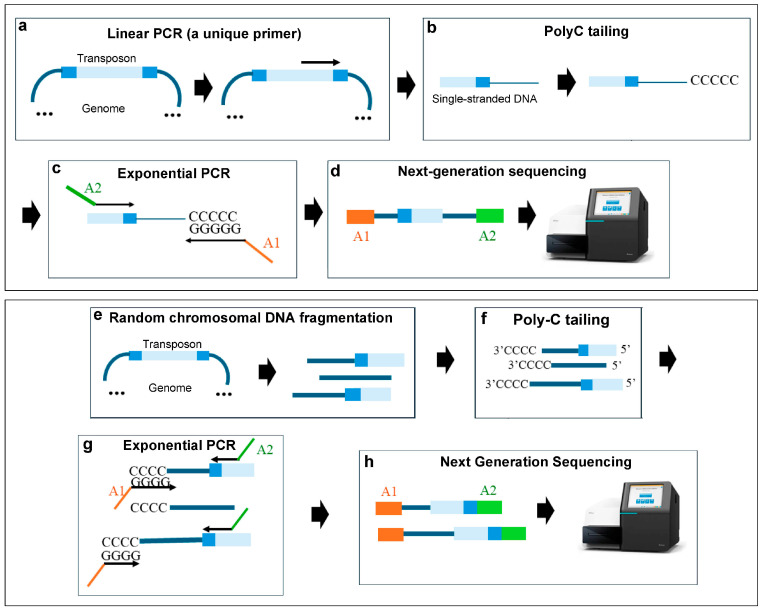
C-tailing method using linear PCR (**a**–**d**) or random fragmentation (**e**–**h**). (**a**) An initial PCR is performed with a single primer specific to the transposon, resulting in single-stranded DNA. (**b**) Cytosine tail is added. (**c**) A second PCR is performed using a poly-G primer that incorporates the adapter 1 sequence and a transposon-specific primer that incorporates the adapter 2 sequence. (**d**) Next-generation sequencing. (**e**) Random chromosomal DNA fragmentation. (**f**) Poly-C tailing. (**g**) PCR using a poly-G primer, a transposon-specific primer. Illumina adapters are incorporated in the primers. (**h**) Next-generation sequencing. The transposon is shown in blue, with its inverted repeats in dark blue; the Illumina adapter 1 sequence (A1) is in orange; the Illumina adapter 2 sequence (A2) is in green.

**Figure 7 ijms-25-11298-f007:**
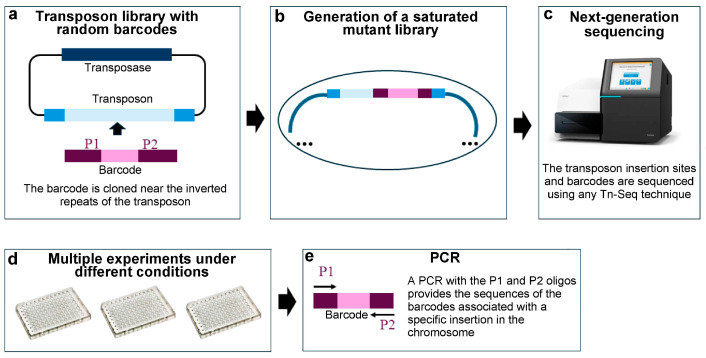
Random barcode transposon sequencing. (**a**) A transposon library with random barcodes is created. (**b**) A saturated mutant library is created using the transposon library harbouring the barcodes. Each gene in which an insertion occurs is associated with a specific barcode. (**c**) After sequencing by Tn-Seq, the barcode for each gene is identified. (**d**,**e**) This mutant library can be used to perform cultures under different conditions and analyse the dynamics of each gene based on its specific barcode. Conditionally essential genes can be identified using this methodology. The transposon is shown in blue, with the inverted repeats in dark blue; the light pink represents the 20 bp random sequence corresponding to the “barcode”; the dark pink represents the sequences for universal primers 1 (P1) and 2 (P2).

**Figure 8 ijms-25-11298-f008:**
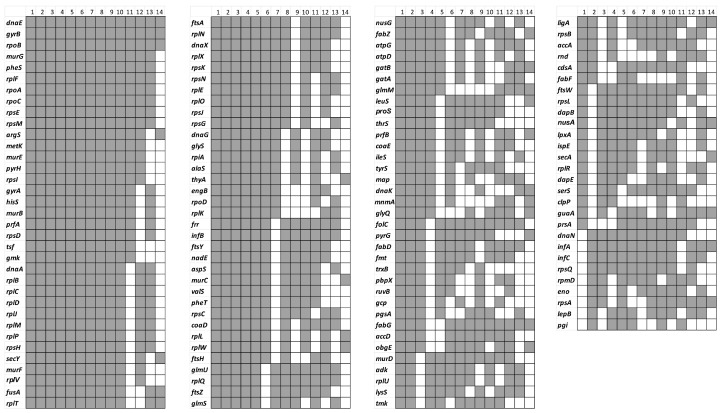
Essential genes identified by Tn-Seq in more than one of the fourteen bacteria shown in Table 1. Grey square indicates that the gene has been identified as essential in at least two bacteria. 1 *Rhodobacter sphaeroides* [17]; 2 *Streptococcus agalactiae* [28]; 3 *Rhodopseudomonas palustris* [31]; 4 *Caulobacter crescentus* [34]; 5 *Azoarcus olearius* [33]; 6 *Porphyromonas gingivalis* [38]; 7 *Herbaspirillum seropedicae* [30]; 8 *E. coli* [35]; 9 *Mycobacterium tuberculosis* [36]; 10 *Salmonella typhimurium* [11]; 11 *Streptococcus pyogenes* [5]; 12 *Streptococcus pneumoniae* [27]; 13 *Ralstonia solanacearum* [26]; 14 *Burkholderia cenocepacia* [39]. The bacteria are arranged based on their number of essential genes, with *Rhodobacter sphaeroides* having the most essential genes in the table and *Burkholderia cenocepacia* the fewest.

**Figure 9 ijms-25-11298-f009:**
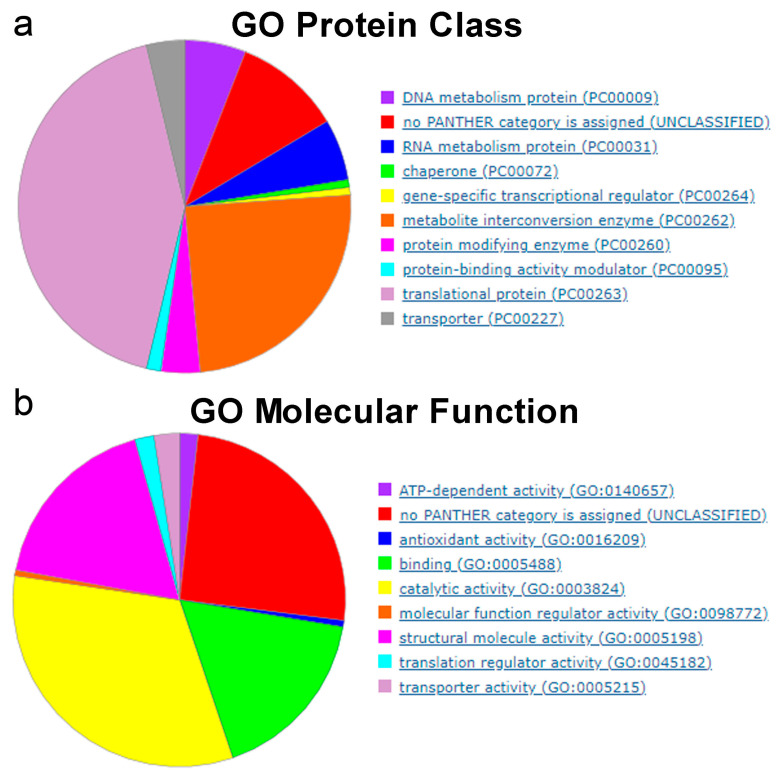
Gene Ontology classification of the essential genes identified by Tn-Seq in the 14 bacterial species shown in Table 1 and Figure 8. (**a**) Protein class. (**b**) Molecular function. Essential genes were classified using *Escherichia coli* as the reference organism (https://geneontology.org/; accessed on 20 September 2024).

**Figure 10 ijms-25-11298-f010:**
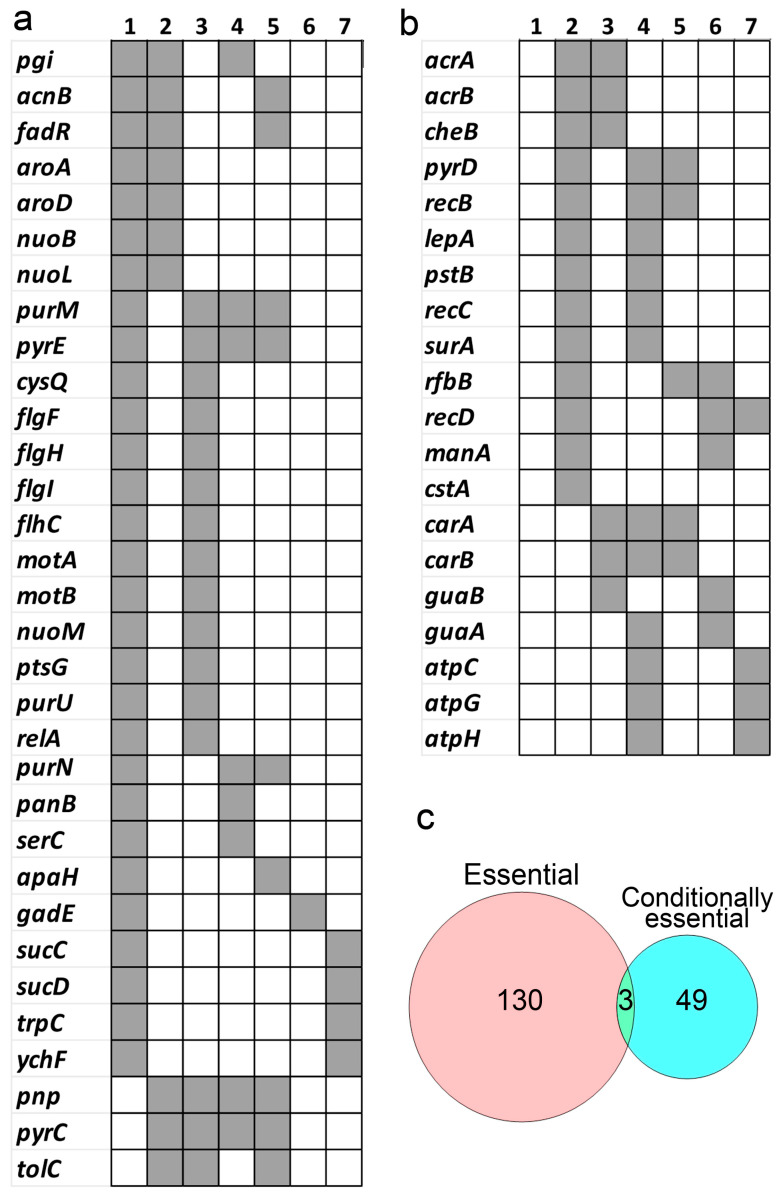
Conditionally essential genes. (**a**,**b**) Conditionally essential genes identified by Tn-Seq in more than one of the eighteen bacteria listed in Table 2. Only 7 of these bacteria share conditionally essential genes with others. The grey square indicates that the gene has been identified as conditionally essential in at least 2 bacteria. 1 *Escherichia coli* [50]; 2 *Salmonella enterica* [52]; 3 *Dickeya dadantii* [44]; 4 *Pseudomonas aeruginosa* [51]; 5 *Vibrio cholerae* [46]; 6 *Mycobacterium tuberculosis* [49]; and 7 *Staphylococcus aureus* [54]. The bacteria are arranged based on their number of conditional essential genes, with *E. coli* having the most essential genes in the table and *S. aureus* the fewest. (**c**) Venn diagram illustrating the limited overlap between the essential genes shown in Figure 8 and the conditionally essential genes shown in Figure 10a,b.

**Figure 11 ijms-25-11298-f011:**
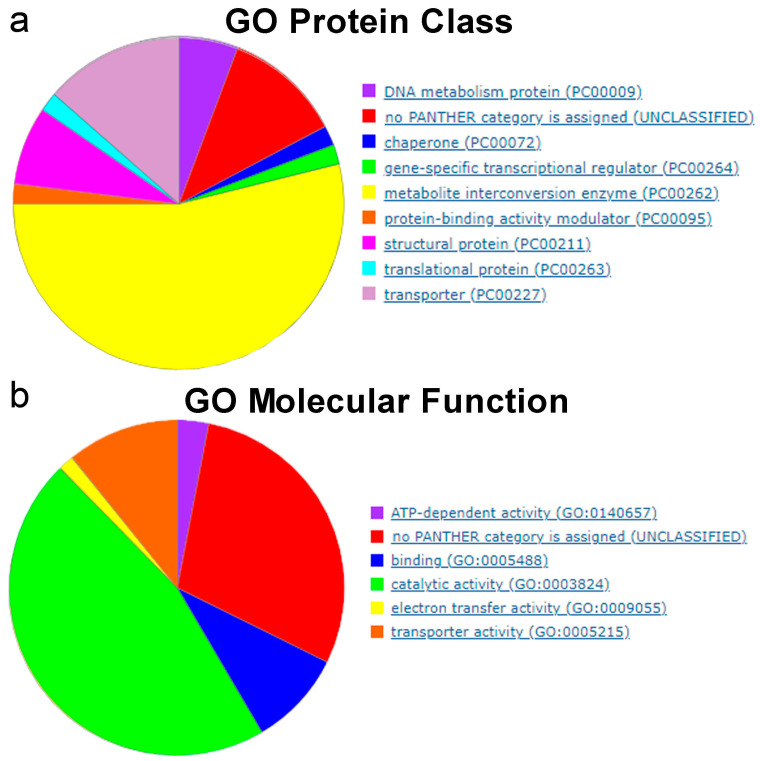
Gene Ontology classification of the conditionally essential genes identified by Tn-Seq in the seven bacteria listed in Figure 10. (**a**) Protein class. (**b**) Molecular function. Conditionally essential genes were classified using *Escherichia coli* as the reference organism (https://geneontology.org/; accessed on 20 September 2024).

## Data Availability

No new data were created or analysed in this review. Data sharing is not applicable to this article, as the original source articles are cited.

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
