# Peer review of "Essential Genes Discovery in Microorganisms by Transposon-Directed Sequencing (Tn-Seq): Experimental Approaches, Major Goals, and Future Perspectives"

_ijms, 2024, doi:10.3390/ijms252011298_

Round 1
Reviewer 1 Report
Comments and Suggestions for Authors
In the present review, the authors present current knowledge and advances on the characterization of essential and conditionally essential genes in 14 bacterial and 3 yeast species. Tn-Seq aims to elucidate the function of each genomic feature and is therefore a critical tool to help interpret the mounting levels of genome sequencing data being generated, thus this is an interesting topic to be published. The manuscript is well-presented and well-written. Tn-Seq methodologies currently used to identify essential and conditionally essential genes are clearly described. The conclusions are consistent with the evidence and they address the main question posed. The future perspectives may be expanded. The references are appropriate. Minor revisions are required before publication. In particular,
- Please use italics for species and gene names throughout the manuscript.
- Please use the term ''species'' when referring to eubacteria (e.g. 14 bacterial species) and yeasts (e.g. 3 yeast species) throughout the manuscript.
- lines 14-15: please replace ''POWERFUL'' with ''powerful''
- line 178: please correct ''S. pneumonieae'' as ''S. pneumoniae''
- line 221: please correct ''Non-circularesed'' as ''Non-circularised''
- line 378: please correct ''Esclerichia'' as ''Escherichia''
- lines 371-372: please replace ''bioinformatics analysis'' with ''bioinformatic analyses''
- line 491: please replace ''thre '' with ''three''
- Figure 9: please denote the data that are presented in (a) and (b)
- line 533: please correct ''Conclussions and future persepctives'' as ''Conclusions and future perspectives''.
- line 551: please replace ''he'' with ''the''
- Please delete the references to Tables and Figures in the ''Conclusions and future perspectives'' section.
- The ''Conclusions and future perspectives'' section may be expanded and improved (please see at: Cain, A.K., Barquist, L., Goodman, A.L. et al. A decade of advances in transposon-insertion sequencing. Nat Rev Genet 21, 526–540 (2020). https://doi.org/10.1038/s41576-020-0244-x).
Author Response
We thank the reviewer for their efforts in reviewing our manuscript and for their comments, which have helped us improve it. To facilitate the review, we have highlighted the changes in yellow.
RESPONSES TO REVIEWER 1’S COMMENTS
“- Please use italics for species and gene names throughout the manuscript.
- Please use the term ''species'' when referring to eubacteria (e.g. 14 bacterial species) and yeasts (e.g. 3 yeast species) throughout the manuscript.
- lines 14-15: please replace ''POWERFUL'' with ''powerful''
- line 178: please correct ''S. pneumonieae'' as ''S. pneumoniae''
- line 221: please correct ''Non-circularesed'' as ''Non-circularised''
- line 378: please correct ''Esclerichia'' as ''Escherichia''
- lines 371-372: please replace ''bioinformatics analysis'' with ''bioinformatic analyses''
- line 491: please replace ''thre '' with ''three''
- Figure 9: please denote the data that are presented in (a) and (b)
- line 533: please correct ''Conclussions and future persepctives'' as ''Conclusions and future perspectives''.
- line 551: please replace ''he'' with ''the''
- Please delete the references to Tables and Figures in the ''Conclusions and future perspectives'' section.”
RESPONSE 1: We apologise for these mistakes that were corrected in the revised manuscript.
COMMENTS 2: “The ''Conclusions and future perspectives'' section may be expanded and improved (please see at: Cain, A.K., Barquist, L., Goodman, A.L. et al. A decade of advances in transposon-insertion sequencing. Nat Rev Genet 21, 526–540 (2020). https://doi.org/10.1038/s41576-020-0244-x).”
RESPONSE 2: As requested, we have expanded the conclussions and future perspectives paragraph (page 19, lines 613-632).
Reviewer 2 Report
Comments and Suggestions for Authors
The review offers valuable insights that are important for bacterial genetics. Although the text contains significant information, its organization needs revision. The flow of information should be cohesive, ensuring that the reader is presented with all the necessary concepts and details. I recommend that the authors include a brief introduction before starting into subitem 1.1, as it currently serves as the opening line of the text. Additionally, the general introduction about transposons should be integrated into the main introduction rather than under subitem 2.1.1.
I) Revise the text for bacterial names, keeping them italicized. Use the full name the first time and the abbreviation afterward.
II)I suggest adding a figure or chart illustrating the transposon structure and related processes would be helpful. The figure should illustrate terminal repeats, the transposase gene, inverted repeats, and flanking sequences, and it should also explain the two main classes of transposons: Class I and Class II.
Line 29: I suggest the authors begin with a brief introduction before starting with subitem 1.1 as the first line of text.
Lines 84-86: This section contains too many items and subitems, which may confuse the reader.
Line 100: Please provide a more precise explanation for the choice of transposon.
Line 108-113: provide a reference for this statement.
Line 469: give a space between “genes in”.
Make sure to leave a space between the words in the lines:
Lines 506 and 508: “genes were”
Line 510: “gene was”
Line 511: “genes are”
Line 513: “genes and”
Comments on the Quality of English Language
In general, English is fine.
Author Response
We thank the reviewer for their efforts in reviewing our manuscript and for their comments, which have helped us improve it. To facilitate the review, we have highlighted the changes in yellow.
RESPONSES TO REVIEWER 2’S COMMENTS
COMMENTS 1: “Although the text contains significant information, its organization needs revision. The flow of information should be cohesive, ensuring that the reader is presented with all the necessary concepts and details. I recommend that the authors include a brief introduction before starting into subitem 1.1, as it currently serves as the opening line of the text. Additionally, the general introduction about transposons should be integrated into the main introduction rather than under subitem 2.1.1.”
RESPONSE 1: We thank the reviewer for this comment. As suggested, we have included a brief introduction on the importance of identifying essential genes (pages 1 and 2, lines 29-59) and have moved section 2.1.1 (“main transposons used for random mutagenesis in microorganisms”) to the Introduction (pages 2 and 3, lines 96-115).
COMMENTS 2: “Revise the text for bacterial names, keeping them italicized. Use the full name the first time and the abbreviation afterward”
RESPONSE 2: The bacterial names have been reviewed and italicised.
COMMENTS 3: “I suggest adding a figure or chart illustrating the transposon structure and related processes would be helpful. The figure should illustrate terminal repeats, the transposase gene, inverted repeats, and flanking sequences, and it should also explain the two main classes of transposons: Class I and Class II”
RESPONSE 3: We thank the reviewer for this suggestion. We have created the requested figure (new Figure 1).
COMMENTS 4: “Line 29: I suggest the authors begin with a brief introduction before starting with subitem 1.1 as the first line of text”.
RESPONSE 4: We have modified the introduction as requested (see Response 3).
COMMENTS 5: “Lines 84-86: This section contains too many items and subitems, which may confuse the reader”.
RESPONSE 5: The reviewer is correct about this point. We have reduced the number of items to improve readability.
COMMENTS 6: “Line 100: Please provide a more precise explanation for the choice of transposon”
RESPONSE 6: We have expanded the explanation for the choice of transposon (see page 3, lines 128-132).
COMMENTS 7: Line 108-113: provide a reference for this statement
RESPONSE 7: A reference has been included (see page 3, line 127).
COMMENTS 8:
“Line 469: give a space between “genes in”.
Make sure to leave a space between the words in the lines:
Lines 506 and 508: “genes were”
Line 510: “gene was”
Line 511: “genes are”
Line 513: “genes and”
RESPONSE 8: We apologise for these mistakes that were corrected in the revised manuscript.
Round 2
Reviewer 2 Report
Comments and Suggestions for Authors
The authors have made substantial improvements by carefully incorporating the suggestions provided. This effort has significantly enhanced the manuscript and will contribute to the field.
Author Response
We thank the reviewer for their constructive comments, which have helped us improve our manuscript.